# The Comet Assay as a Tool to Detect the Genotoxic Potential of Nanomaterials

**DOI:** 10.3390/nano9101385

**Published:** 2019-09-27

**Authors:** Alba García-Rodríguez, Laura Rubio, Laura Vila, Noel Xamena, Antonia Velázquez, Ricard Marcos, Alba Hernández

**Affiliations:** 1Department of Genetics and Microbiology, Faculty of Biosciences, Universitat Autònoma de Barcelona, 08193 Cerdanyola del Vallès (Barcelona), Spain; albagr.garcia@gmail.com (A.G.-R.); lauravilavecilla@hotmail.com (L.V.); noel.xamena@uab.cat (N.X.); antonia.velazquez@uab.cat (A.V.); 2Nanobiology Laboratory, Department of Natural and Exact Sciences, Pontificia Universidad Católica Madre y Maestra, PUCMM, Santiago de los Caballeros 50000, Dominican Republic; l.rubio@ce.pucmm.edu.do; 3Consortium for Biomedical Research in Epidemiology and Public Health (CIBERESP), Carlos III Institute of Health, 28029 Madrid, Spain

**Keywords:** comet assay, FPG enzyme, TiO_2_NP, SiO_2_NP, ZnONP, CeO_2_NP, AgNP, multi-walled carbon nanotubes (MWCNT)

## Abstract

The interesting physicochemical characteristics of nanomaterials (NMs) has brought about their increasing use and, consequently, their increasing presence in the environment. As emergent contaminants, there is an urgent need for new data about their potential side-effects on human health. Among their potential effects, the potential for DNA damage is of paramount relevance. Thus, in the context of the EU project NANoREG, the establishment of common robust protocols for detecting genotoxicity of NMs became an important aim. One of the developed protocols refers to the use of the comet assay, as a tool to detect the induction of DNA strand breaks. In this study, eight different NMs—TiO_2_NP (2), SiO_2_NP (2), ZnONP, CeO_2_NP, AgNP, and multi-walled carbon nanotubes (MWCNT)—were tested using two different human lung epithelial cell lines (A549 and BEAS-2B). The comet assay was carried out with and without the use of the formamidopyrimidine glycosylase (FPG) enzyme to detect the induction of oxidatively damaged DNA bases. As a high throughput approach, we have used GelBond films (GBF) instead of glass slides, allowing the fitting of 48 microgels on the same GBF. The results confirmed the suitability of the comet assay as a powerful tool to detect the genotoxic potential of NMs. Specifically, our results indicate that most of the selected nanomaterials showed mild to significant genotoxic effects, at least in the A549 cell line, reflecting the relevance of the cell line used to determine the genotoxic ability of a defined NM.

## 1. Introduction

Nanomaterials (NMs) are being increasingly used in many fields, due to their new physicochemical properties at the nanometric scale [1]. In this context, the evidence seems to indicate that the use of such NMs will continue experiencing an exponential increase. This opens a new scenario where people will certainly be exposed to such NMs. Since human health risks associated with such exposures are uncertain, new information on their potentially harmful effects is urgently required.

Among the different health effects that NMs can cause, those related to their potential interaction with DNA requires special attention. Thus, the detection of the genotoxic effects induced by NMs is emerging as a specific research field covering such demands [2,3]. Since different NMs can react with DNA following different mechanisms, a wide range of tools have been proposed to detect and quantify different genotoxic effects; among them, the comet assay stands out.

The comet assay measures the induction of DNA breaks as well as oxidatively damaged DNA bases by using single-cell gel electrophoresis [4,5]. The simplicity of the assay and its potential use in any type of eukaryotic cell has expanded its use in many fields, and this assay is successfully used in in vitro and in vivo testing, including human biomonitoring studies [6]. The comet assay has already been used to test different NMs as it has been summarized in different reviews [7,8,9]. Several criticisms have been raised about the use of the comet assay to test NMs arguing that their presence in the head, or in the tail of the comet, can interfere with the scoring [7]. This means that the scoring of NMs using the comet assay is under discussion. At present, there is no official guideline for using the comet assay in in vitro studies. In this frame, the EU project NANoREG has aimed to establish strong protocols to be used in the testing of NMs, mainly from a regulatory point of view. According to its simplicity, sensitivity, and speed, the comet assay was one of the proposed test protocols. To reach that objective, we here present the results obtained with the comet assay testing eight different NMs namely NM100 and NM101 (TiO_2_NPs), NM110 (ZnONPs), NM200 and NM203 (SiO_2_NPs), NM212 (CeO_2_NPs), NM300K (AgNPs), and NM401 (MWCNT, multiwalled carbon nanotubes). Taking into account that inhalation is one of the main routes of exposure to NMs, two different human cell lines (A549 and BEAS-2B) have been used as a model of human bronchial epithelial cells. To cover one of the aims of the NANoREG project a high throughput approach was used. Thus, GelBond films (GBF) instead of glass slides were used, allowing the fitting of 48 microgels on the same GBF, in comparison with the 1–2 samples contained in a glass slide.

## 2. Material and Methods

### 2.1. Selected NMs

The tested NMs were: 100 and 6 nm anatase TiO_2_NPs (NM100 and NM101), 18.3 and 24.7 nm amorphous SiO_2_NPs (NM200 and NM203), 147 nm ZnONPs (NM110), 33 nm CeO_2_NPs (NM212), AgNPs (NM300K), and 64.2 nm MWCNT (NM401). All eight NMs were supplied by the NANoREG Consortium, and the EU Joint Research Centre at Ispra (Italy) prepared and sent aliquots to all the participants. Although these NMs were well characterized in the frame of the project, we confirmed nanoparticle size and morphology by transmission electron microscopy (TEM) using a JEOL 1400 instrument (JEOL Ltd., Tokyo, Japan). In addition, the hydrodynamic size in whole-cell culture medium was measured by using dynamic light scattering (DLS). This analysis was performed on a Malvern Zetasizer Nano-ZS zen3600 (Malvern Panalytical, Malvern, UK) instrument.

### 2.2. Cell Culture

The transformed human bronchial epithelial BEAS-2B cell line was kindly provided by Dr. H. Karlson, from the Swedish Karolinska Institute (Stockholm, Sweden). The adenocarcinomic human alveolar basal epithelial A540 cell line was kindly provided by Dr. G. Linsel, from the Federal Institute for Occupational Safety and Health of Germany (BAuA, Berlin, Germany), BEAS-2B cells were cultured as a monolayer in 75 cm^2^ culture flasks coated with 0.03% collagen in serum-free bronchial epithelial cell growth medium (BEGM, Lonza, CA, USA) and passaged weekly. A549 cells were also cultured in 75 cm^2^ culture flasks with DMEM high glucose (Life Technologies, Carlsbad, CA, USA) supplemented with 10% fetal bovine serum (FBS; PAA, Pasching, Austria), 1% of non-essential amino acids (NEAA; PAA, Pasching, Austria) and 2.5 μg/mL Plasmocin™ (InvivoGen, San Diego, CA, USA). Log-phase cells were grown on 12 well plates without 0.03% collagen coating, to determine, cytotoxicity and DNA damage levels. Cells were maintained in a humidified atmosphere of 5% CO_2_ and 95% air at 37 °C.

We selected epithelial lung cells because inhalation is considered one of the most relevant exposure routes for humans. In addition, both BEAS-2B cells have been extensively used in the testing of nanomaterials.

### 2.3. Cell Viability

BEAS-2B and A549 cells were exposed to different concentrations of the selected NMs, according to previous toxicity data, and exposures lasting for 24 h. The highest tested dose ranged from 10 μg/mL, for the most toxic NMs (ZnONP), to 200 μg/mL for the least toxic NMs. According to the obtained toxicities, up to 4 different concentrations were selected to determine cell viability. The 4 appropriate subtoxic concentrations were chosen to work with the comet assay enabling the establishment of dose/response relationships. Untreated cells, just with cell culture medium, were used as a negative control.

Toxicity was measured directly counting the number of cells surviving the 24 h exposure treatments. The procedure was as follows: after exposure, cells were washed three times with 0.5 mL of PBS (1%), to eliminate dead cells, and incubated 3 min at 37 °C with 0.25 mL of trypsin-EDTA (Ethylenediaminetetraacetic acid) (1%) to detach and individualize them. After these initial steps, cells were diluted (1/10) in ISOTON™ (isotonic buffer) and counted with a ZTM Series colter-counter (Beckman Coulter Inc., Brea, CA, USA) [10].

The final viability values were determined by averaging three independent viability experiments, each containing three replicates per each concentration.

### 2.4. The Comet Assay

The alkaline comet assay facilitates the determining of the levels of both the genotoxic (DNA breaks) and the oxidative DNA damage (ODD). The detection of ODD can be easily detected when the comet assay is complemented with the use of formamidopyrimidine DNA glycosylase (FPG) enzyme. This enzyme detects oxidized DNA bases, cut them and, consequently, transient DNA breaks are generated. The net oxidative effect is determined by subtracting the breaks induced in normal conditions from those obtained when FPG enzyme is used. The used protocol is as follows: exposed/control cells were washed with PBS 1X thrice, detached with trypsin-EDTA 1%, incubated at 37 °C for 5 min, and centrifuged at 130 g for 8 min. After that, cells were resuspended in PBS 1X to obtain the concentration of 1 × 10^6^ cells/mL. The obtained cells were mixed 1:10 with 0.75% low melting point agarose at 37 °C, dropped on GelBond^®^ films (GBF) (Life Sciences, Vilnius, Lithuania) in triplicates, and lysed in cold lysis buffer overnight at 4 °C. As a high throughput approach, the use of GBF instead of glass slides allowed for the fitting of 48 microgels on the same GBF.

After that, the GBF were gently washed twice with enzyme buffer for 5 and 50 min respectively at 4 °C. Then, the GBF were incubated with enzyme buffer containing, or not, FPG enzyme 1:25,000 for 30 min at 37 °C, followed by a washing step with electrophoresis buffer for 5 min. A second incubation in electrophoresis buffer was followed, to allow DNA unwinding and expression of alkali-labile sites, for 25 min at 4 °C. Subsequently, electrophoresis was carried out at 20 V and 300 mA at 4 °C. GBF were washed twice with cold PBS 1X for 5 and 10 min respectively, cells were fixed in absolute ethanol for at least 1 h and air-dried overnight at room temperature. Cells were stained with 1:10,000 SYBR Gold in TE buffer for 20 min at room temperature. GBF were mounted and visualized in an epifluorescent microscope (Olympus BX50, Hamburg, Germany) at 20× magnification. The DNA damage was quantified with the Komet 5.5 Image analysis system (Kinetic Imaging Ltd, Liverpool, UK) as the percentage of DNA in the tail. A total of 100 comet images randomly selected were analyzed per sample. Two different samples were analyzed for each condition, in each one of the three experiments performed.

### 2.5. Statistical Analysis

All measurements were made in triplicate, at least for two separate experiments. Results are expressed in mean ± standard error. The one-way ANOVA with Tukey’s post-test and an unpaired Student’s *t*-test were used to compare differences between means. Data were analyzed with GraphPad Prism version 5.00 for Windows (GraphPad Software, San Diego, CA, USA, http://graphpad.com). Differences between means were considered significant at *p* < 0.05.

## 3. Results and Discussion

### 3.1. Nanoparticles Characterization

First of all, we wish to point out that the nanomaterials used were supplied by the UE JRC at Ispra, and are considered as reference nanomaterials, and used in many different EU projects. This means that they have been extensively characterized. To gather sound genotoxicity data, it is necessary to ensure that nanoparticles are well dispersed [11]. In our case, we have used the dispersion protocol generated in the frame of the NanoGenotox EU project [12]. As observed in Figure 1, TEM images show that each NMs is dispersed distinctly, depending on its primary structure. For example, AgNPs and TiO_2_NPs (NM101) are clearly not or less agglomerated than the rest of the NPs. Then, by selecting over 100 particles in random fields of view, we determined the TEM diameters in dry conditions, and the obtained results are indicated in Table 1. The same table also shows the DLS results, that measured the average of the NPs hydrodynamic size when were dispersed and suspended in a liquid. As expected, the average of the hydrodynamic sizes, once dispersed in the culture medium, are higher than those reported by TEM in the dried form, due to their interactions with the proteins. Since exposures last for 24 h, we aimed to detect if time has important effects on dispersion by inducing NMs aggregation. Thus, our results summarized in Table 1 do not seem to indicate that the incubation time has a direct effect on the NMs agglomeration. Concretely, only NM200, NM203, and NM300K showed apparent increases in the hydrodynamic size after 24 h, regarding the values observed just after the dispersion procedure (0 h). It should be indicated that, due to the fibrillary characteristics of MWCNT NM401, we do not report DLS values for this NM (ND). Nevertheless, we indicated their average length and thickness, as evaluated by TEM. Although some of the values, observed mainly after 48 h, were difficult to interpret, these discrepancies over-time are not important, because only exposures lasting for 24 h were used in the comet assay. 

### 3.2. Toxic/Genotoxic Effects of NM100 and NM101 (TiO_2_NPs)

TiO_2_NPs are one of the most produced NMs. They are included in many products, such as paints, coatings, plastics, papers, inks, medicines, pharmaceuticals, food products, cosmetics, sunscreens, and toothpaste [13]. This extensive use makes it necessary to evaluate their potential risk for human health.

Toxicity data are required to determine the range of concentrations to be used in genotoxicity studies. In our case, we have not detected relevant toxic effects for the two selected TiO_2_NPs (Figure 2A and Figure 3A). Although TiO_2_NPs have largely been evaluated from the genotoxic point of view, the already reported results are not sufficiently clear to reach a convincing conclusion [14]. Nevertheless, in our study, we have demonstrated that both TiO_2_NPs (NM100 and NM101) are able to induce direct DNA strand breaks, as detected by the comet assay (Figure 2 and Figure 3), although no oxidative DNA damage induction was detected on the DNA bases. Overall, we can say that A540 cells are more sensitive than BEAS-2B cells and, consequently, exposures induced higher levels of damage in that cell line. Furthermore, NM100 showed to be slightly more genotoxic than NM101. From the literature, positive DNA damage induction for both used cell lines has been reported. Nevertheless, some studies failed to detect DNA damage induction. Thus, negative results in BEAS-2B cells were obtained under different exposure scenarios [15,16,17]. In addition, negative findings in the comet assay were also reported in the A549 cells [18]. From the different positive reports in BEAS-2B cells, nanosized anatase and fine rutile produced a concentration and time-dependent effect in exposures lasting up to 72 h [19]. On the other hand, using the NM100 and NM101 nanomaterials (as in our study) Di Bucchiano et al. demonstrated a weak but positive induction of genotoxic effects, NM100 being the most dangerous [20]. A recent study in BEAS-2B indicated that the induced effects were associated with the shape. In this way, only those forms that were clearly internalized (food grade, P25, and platelets) were able to induce genotoxicity [21]. As in our case, A549 cells showed a higher sensitivity to TiO_2_NPs and, consequently, positive effects were reported for both small and spherical anatase and rutile forms after short-term exposures [22], and also after long-term exposures lasting up to two months [23]. In addition, the positive induction of DNA damage was reported at non-toxic concentrations, but induced DNA damage decreased when exposures extended from 3 to 24 h [24]. Interestingly, there is one study comparing the effects of TiO_2_NPs in BEAS-2B and A549 cells. In that study, direct and oxidative DNA damage was only observed in A549 cells [16], confirming the highest sensitivity of this cell line.

ZnONPs, together with TiO_2_NPs, are among the most produced NMs. ZnONPs are widely used in personal care products, sensors, antibacterial creams, and biomedical applications. Their broad range of applications raises concerns regarding their potential health effects [25]. In our study, ZnONPs (NM110) exerted a very important toxic effect in both cell lines, with the concentration of 10 µg/mL being completely toxic to BEAS-2B cells. However, ZnONPs were unable to cause direct DNA damage in A549 and BEAS-2B cells. It should be indicated that some genotoxic effects were observed in A549 cells when the FPG enzyme was used. Thus, significant increases in the levels of oxidative DNA damage were observed in cells exposed to 0.2 and 6 µg/mL, as indicated in Figure 4.

ZnONPs are easily taken up by cells, and quickly dissolved intracellularly to their ionic form, which can cause the strongly observed toxicity. The use of ICP-MS techniques demonstrated that, after 48 h in culture media, more than 80% of the initial ZnONPs dissolved to their ionic form [26]. In that study, although the authors found that exposures lasting for 24 h induced significant levels of DNA damage in mouse fibroblast cells, this DNA damage induction was not evident when cells were exposed long-term (up 12 weeks). Similar effects were observed in BEAS-2B cells, where short-term exposures lasting for 3–6 h were able to induce DNA damage in the absence of BSA [27]. Positive genotoxic effects were also obtained in A549 cells at short exposure times (4 h). Nevertheless, this genotoxicity was not associated with oxidative DNA damage, nor the induction of intracellular Reactive Oxygen Species (ROS) levels [28]. More recently, ZnONPs were also evaluated in A549 cells affecting cell survival, and inducing high levels of DNA damage at cytotoxic concentrations. Lower concentrations also were able to induce a genotoxic response and the induced damage persisted over 24 h [24]. The disparity of reported data, mainly related to the exposure times, reflects the relevance of the intracellular solubility as a modulator of the observed effects.

Due to their appealing properties, SiO_2_NPs are extensive and increasingly used in agriculture, food, and consumer products including cosmetics. Accordingly, large amounts are placed in the global market and, consequently, into the environment. Many products containing SiO_2_NPs are listed in a consumer product inventory and placed in the third position among the most produced NMs worldwide [20]. Although many studies have been carried out, to identify the toxicological mechanisms of action of SiO_2_NPs, no conclusive positions have been established linking their physicochemical properties to toxicity, bioavailability, or human health effects [29]. In a similar way, although potential biomarkers of genotoxicity have been suggested, the obtained experimental results are not conclusive enough, due to a variety of factors [30]. Our findings showed similar results for both evaluated SiO_2_NPs, with BEAS-2B being more sensitive than A549 cells to their toxic effects. Regarding their genotoxic potential, both SiO_2_NPs were able to induce direct genotoxicity in both cell lines. Nevertheless, in A549 cells both SiO_2_NPs were able to induce oxidative DNA damage at the concentrations of 15 and 30 µg/mL (Figure 5 and Figure 6).

No studies using BEAS-2B cells have been found testing the genotoxicity of SiO_2_NPs and using the comet assay. Nevertheless, two studies reported data using the A549 cell line. In the first study, short periods of incubation (15 min and 4 h) did not induce DNA strand breaks or FPG sensitive sites, when testing two different SiO_2_NPs sizes (16 and 60 nm) [31]. The second study evaluated the effects of non-cytotoxic concentration of SiO_2_NPs, with an average diameter of 39.6 nm, in treatments lasting for 24 h. Results indicated that the exposure did not increase the intracellular levels of ROS or the primary DNA damage, as measured with the comet assay [32]. Interestingly, SiO_2_NPs exerted a synergistic action on the effects of lead, amplifying the levels of ROS and DNA damage induced by lead [32].

Regarding CeO_2_NPs, they have widespread use in industry, cosmetic and consumer products. Its use as automobile exhaust catalysts draws special attention, due to their extensive release into the environment. At present, there is a growing interest in this compound, since it has been proposed for biomedical use, due to the potent regenerative antioxidant properties. Although in vivo exposures to CeO_2_NPs can exert respiratory tract adverse effects, such as sensory irritation and airflow limitation [33], its analogous activity to two key antioxidant enzymes, such as superoxide dismutase and catalase, explains its role as a potent free radical scavenger. Among the proposed health effects, the potential application in neurodegenerative pathologies stands out [34]. 

Our results indicate that CeO_2_NPs (NM212) does not exert any toxic effect in the used cell lines. In addition, non-genotoxic effects were observed in both A549 and BEAS-2B (Figure 7). It is interesting to note the observed reduction of oxidative DNA damage observed in treated A549 cells, which would support the antioxidant potential of CeO_2_NPs.

Although some authors have reported the genotoxic effects of CeO_2_NPs, there is no general agreement about this point [35]. It has been proposed that the apparent discordant reported data are the result of both the pro-oxidant and anti-oxidant role of CeO_2_NPs. Nevertheless, in a recent study using a wide set of cell lines, from both tumoral and non-tumoral origin, only anti-oxidant effects were observed [36]. Regarding the use of BEAS-2B and A549 cells to evaluate the genotoxic effects of CeO_2_NPs, only three studies have been found in the literature using the comet assay. In BEAS-2B cells, no induction of intracellular ROS of DNA damage was observed [37], but positive induction of DNA damage was observed in A549 cells [24]. In that study, the effects were mainly observed after short exposures (3 h), while only a slight effect at the highest tested dose (42 µg/mL) was observed. Conversely, Frieke-Kuper et al. used both cell lines and demonstrated that in both cases exposures lasting for 24 h of CeO_2_NPs induced significant levels of DNA damage, as evaluated using the comet assay [38]. Nevertheless, the use of an in vitro 3D human bronchial epithelial model indicated that the mucociliary defense prevents CeO_2_NPs from reaching the respiratory epithelial cells [38].

The wide spectrum of applications of AgNPs in biomedicine, and in related fields, explain their extended use [39]. Due to their proved antimicrobial efficacy, AgNPs are incorporated into different materials such as textile fibers and wound dressings. This explains that, although the production of AgNPs is significantly lower than that of other NMs, they constitute the most popular advertised nanomaterial in the Nanotechnology Consumer Products Inventory [40]. Among the different harmful effects associated with AgNPs exposure, it has been extensively reported that may cause genotoxicity, although additional data are required to assess its carcinogenic potential [41].

In our study, AgNPs induced significant cytotoxicity in both cell lines, BEAS-2B cells showing a higher sensitivity (Figure 8). In addition, AgNPs exposure was also able to induce genotoxic damage in both cell lines. Nevertheless, when the FPG enzyme was used to detect the induction of oxidatively damaged DNA, only A549 cells showed this type of damage (Figure 8). Until now, four studies have been published testing AgNPs in BEAS-2B using the comet assay [42,43,44,45]. Interestingly, completely different genotoxicity data were reported in the indicated studies. Thus, AgNPs with a size ranging from 20–200 nm, were unable to induce DNA strand breaks in treatments lasting for 4 h [44]. Contrarily, positive DNA strand breaks were observed using different types of AgNPs as citrate coated (10, 40 and 75 nm), PVP coated (10 nm), and uncoated (50 nm). Positive induction of primary DNA damage was observed for all forms, but only after treatments lasting for 24 h. In contrast, negative results were obtained for short exposures (4 h), and no induction of intracellular ROS was detected, assuming that the observed DNA damage was not a as result of the oxidative stress status [45]. These results are opposed to those reported by Nymark et al., who indicated DNA strand breaks, together with intracellular ROS induction, when BEAS-2B cells were exposed to PVP-coated AgNPs (average diameter 42.5 ± 14.5 nm) for treatments lasting 4 and 24 h [43]. The genotoxic potential of AgNPs using A549 cells has also reported variability of data, with negative results using nanoparticles sized 20–200 nm and exposures lasting for 4 h [44], and negative results in PVK-coated AgNPs sized 20 nm, although small sizes (20 nm) induced a significant induction of DNA damage [46]. This positive induction was also observed in cells exposed for 24 h to 20–50 nm diameter nanoparticles [47]. The size of AgNPs has been considered as one of the characteristics modulating genotoxicity. High levels of DNA damage were observed for the smaller sized AgNPs (50 nm) when three different sizes (50, 80 and 200 nm) were evaluated [48]. Similar results were reported when 20 and 200 nm were compared. In that case, higher levels of DNA damage were observed when cells were exposed to 20 nm AgNPs, in exposures lasting for 2 and 24 h. The use of the FPG enzyme demonstrated that most of the damage was caused by oxidation at DNA bases [49]. Finally, it should be indicated that, using the same type of AgNPs as in our study (NM300K), El Yamani et al. reported positive induction of DNA damage, independently of the exposure time (3 or 24 h), where the observed damage was mainly due to oxidative damage, as measured combining the comet assay with the use of FPG enzyme [24].

MWCNTs are fibrous materials formed from honeycomb crystal lattice layers of graphite wrapped into a multiple tube shape. They are applied in multiple fields, including in their use as semiconductors, solar cell mobiles, and optical instruments. From the toxicological point of view, they are considered to have carcinogenic potential, causing lung tumors [50]. However, the underlying mechanisms are not well understood and, in particular, the reported genotoxicity data are conflicting [51]. 

In our study, MWCNT (NM401) did not exert toxic effects on A549 cells, but toxicities around 50% were observed in the BEAS-2B cell exposed to the whole range of tested concentrations (Figure 9A). Regarding the ability of this NM to induce DNA damage, our results indicate that direct genotoxicity was not observed at any of the tested concentrations, in any of the used cell lines. Nevertheless, oxidative DNA damage was observed in the A549 cell line, but only at the lowest concentrations (15 and 30 µg/mL) (Figure 9). Using BEAS-2B cells, the literature mainly reports negative results [52,53,54]. Although the comparison of straight and tangled MWCNTs demonstrated that the straight forms were positive at low concentrations, tangled MWCNTs showed to be only positive at the highest tested dose [55]. The A549 cell line has also been used in studies testing the genotoxic potential of MWCNTs in the comet assay, with contradictory results. Negative results were reported in exposures lasting for 24–72 h [54,56], although a positive response in the comet assay without FPG was also observed [57,58]. Nevertheless, both studies got negative results when the comet assay was complemented with the use of the FPG enzyme. The role of modifications in the MWCNTs structure has been evaluated, comparing the pristine versus the functionalized forms. Positive effects were reported, independently if the pristine from was compared with the OH-functionalized form [58] or with the acid-treated forms [59]. However, when A549 cells were used to test pristine versus carboxyl MWCNTs, only positive results were obtained in the comet assay. Interestingly, both MWCNTs forms gave positive induction of strand breaks when BEAS-2B cells were used [53].

## 4. Conclusions

A summary of the results reported above is shown in Figure 10. From the reported data, we can conclude that the comet assay is a sensitive tool to determine the ability of NMs to produce DNA damage. According to the obtained results, the tested nanomaterials can be classified as weak genotoxic agents and, in such cases, most of the obtained results do not follow a clear dose–response pattern. This can result from the inherent variability associated with sampling size, an important factor when the increases in the levels of genetic damage are small. Alternatively, it is feasible that the cell uptake of nanomaterials is not always associated with their concentration in the culture media. At high concentrations, aggregation can occur, reducing the cell uptake. Furthermore, the observed differences between cell lines indicate that the selection of the cell-type is an important factor when the comet assay is used for testing NMs. In our case, we have demonstrated the highest sensitivity of the A549 cell line, regarding BEAS-2B cells. This is especially true when the induction of oxidative damage on DNA bases was evaluated. Thus, the positive induction of oxidative damage in A549 cells exposed to ZnONPs, SiO_2_NPs, AgNPs, and MWCNT was not obtained in BEAS-2B cells. This could be due to the important antioxidant capacity of this cell line [60]. In fact, different behaviors of both cell lines have also been observed by other authors regarding different genetic endpoints. Thus, A549 cells were more sensitive when exposed to cadmium than BEAS-2B cells, the higher sensitivity of A549 cells being associated to changes in cell type-specific gene expression patterns, including the induction of genes coding for metallothioneins, the oxidative stress response, cell cycle control, mitotic signaling, apoptosis and DNA repair pathways [61]. Similarly, A549 cells were more sensitive when exposed to TiO_2_NPs than BEAS-2B cells, and this sensitivity was associated with changes in the expression of DNA repair genes [62]. Finally, the use of hydrophobic plastic supports (HBF) permits an easy throughput approach of the comet assay and, in addition, the use of FPG enzyme has become a powerful tool to understand the mechanism of action of the tested nanomaterials, detecting those inducing oxidative DNA damage on the DNA bases.

## Figures and Tables

**Figure 1 nanomaterials-09-01385-f001:**
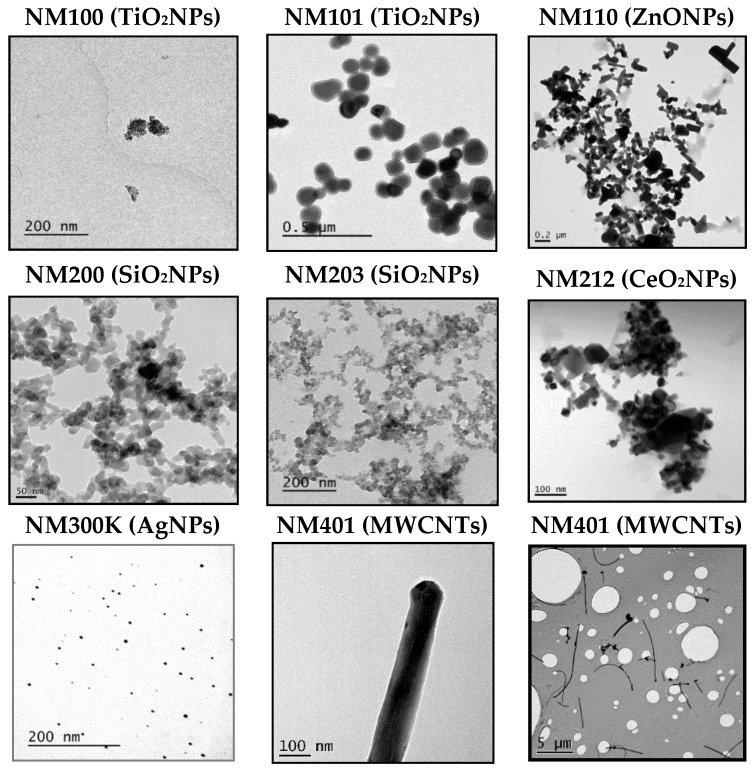
Representative TEM images of the eight used nanoparticles (NPs). Figures correspond to TiO_2_NPs (NM100), TiO_2_NPs (NM101), ZnONPs (NM110), SiO_2_NPs (NM200), SiO_2_NPs (NM203), CeO_2_NPs (NM212), AgNPs (NM300K), and MWCNT (NM401). For NM401 two images are include to visualize their length and thickness.

**Figure 2 nanomaterials-09-01385-f002:**
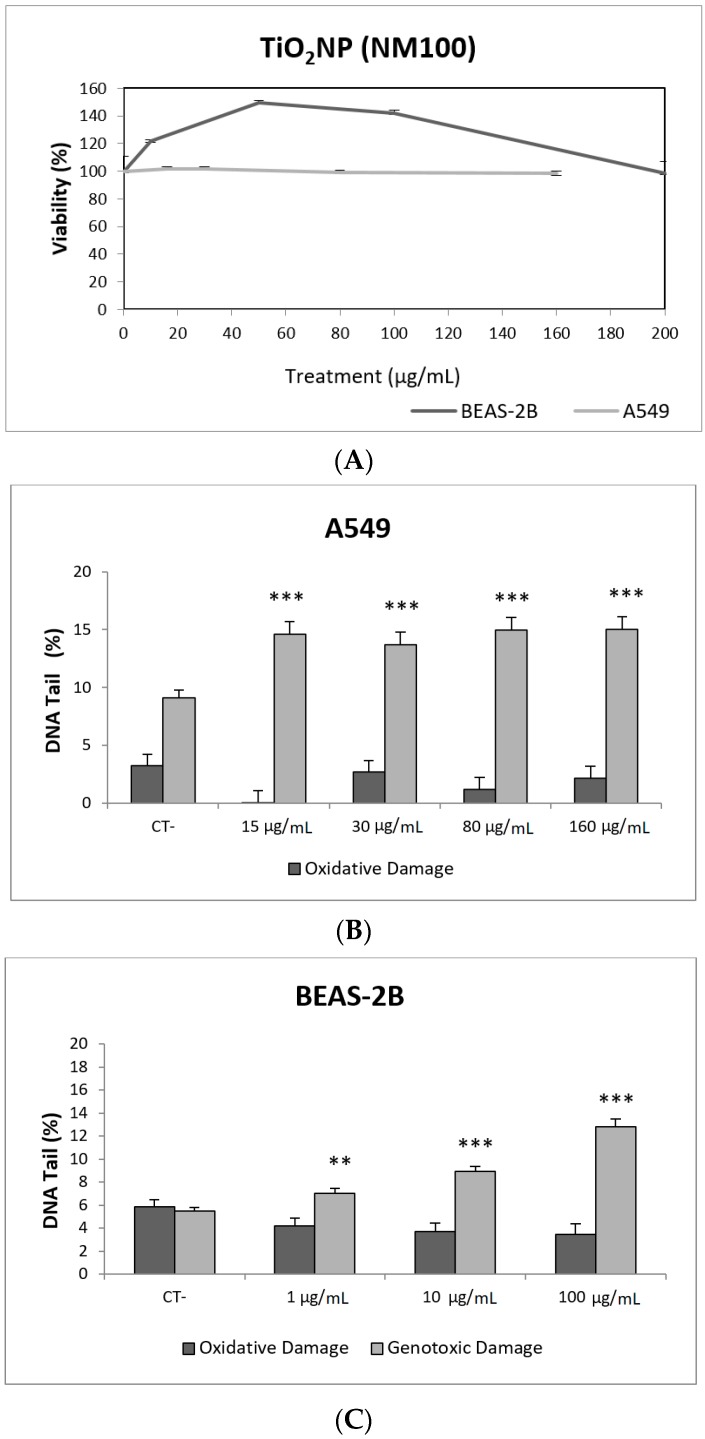
Results from the TiO_2_NPs (NM100) on A549 and BEAS-2B cells. (**A**) Toxicity curves for both cell lines. (**B**) Genotoxicity results obtained in A549 cells. (**C**) Genotoxicity results obtained in BEAS-2B cells. Genotoxicity data are plotted as mean ± SEM of two independent experiments. ** *p* ≤ 0.01, *** *p* ≤ 0.001 (one way-ANOVA). CT, control.

**Figure 3 nanomaterials-09-01385-f003:**
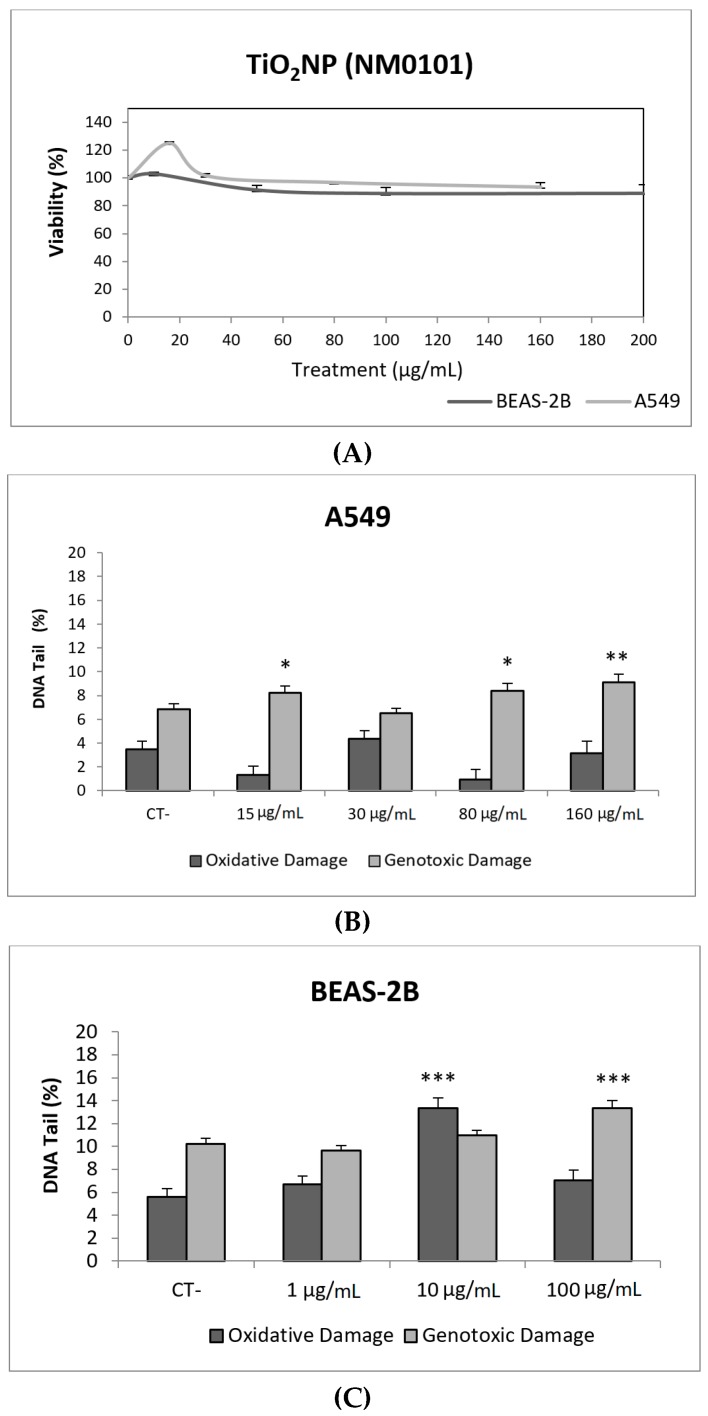
Results from the TiO_2_NPs (NM101) on A549 and BEAS-2B cells. (**A**) Toxicity curves for both cell lines. (**B**) Genotoxicity results obtained in A549 cells. (**C**) Genotoxicity results obtained in BEAS-2B cells. Genotoxicity data are plotted as mean ± SEM of two independent experiments. * *p* ≤ 0.05, ** *p* ≤ 0.01, *** *p* ≤ 0.001 (one way-ANOVA). CT, control.

**Figure 4 nanomaterials-09-01385-f004:**
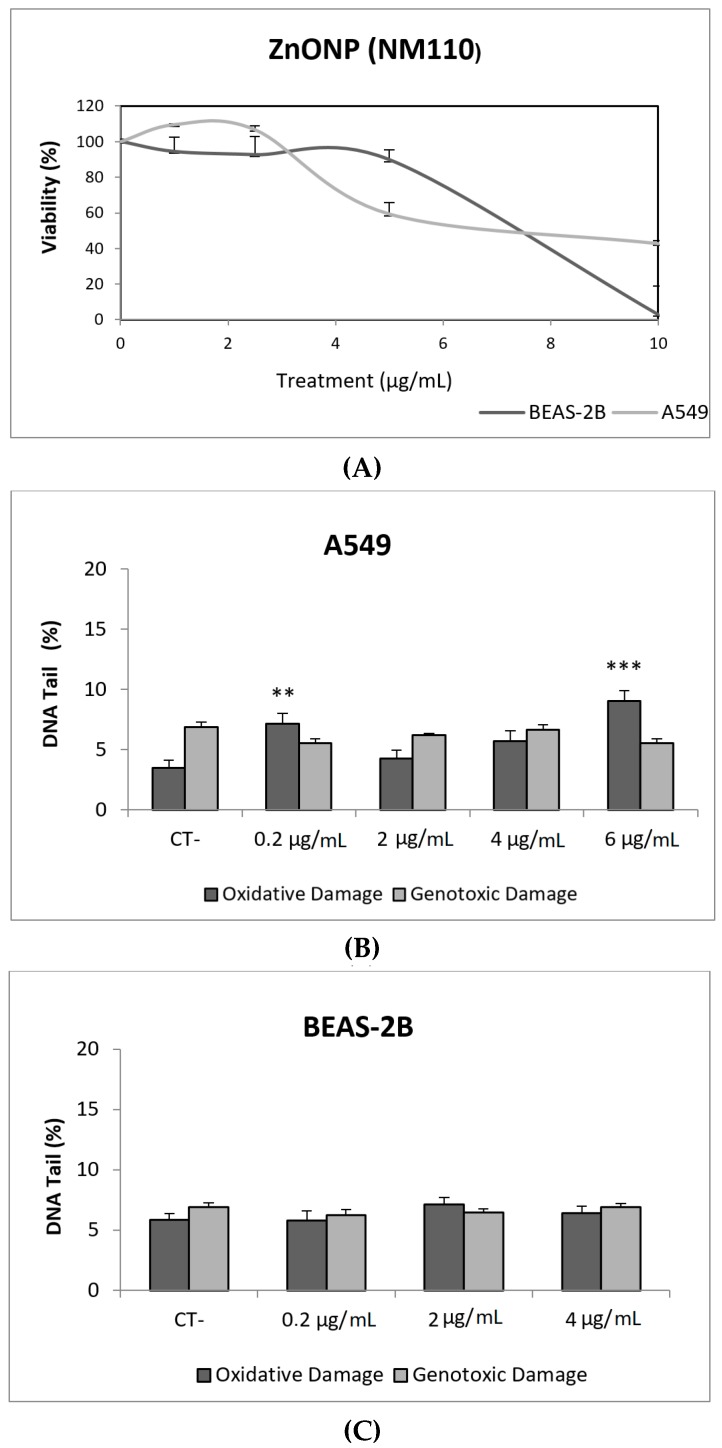
Results from the ZnONPs (NM110) on A549 and BEAS-2B cells. (**A**) Toxicity curves for both cell lines. (**B**) Genotoxicity results obtained in A549 cells. (**C**) Genotoxicity results obtained in BEAS-2B cells. Genotoxicity data are plotted as mean ± SEM of two independent experiments. ** *p* ≤ 0.01, *** *p* ≤ 0.001 (one way-ANOVA). CT, control.

**Figure 5 nanomaterials-09-01385-f005:**
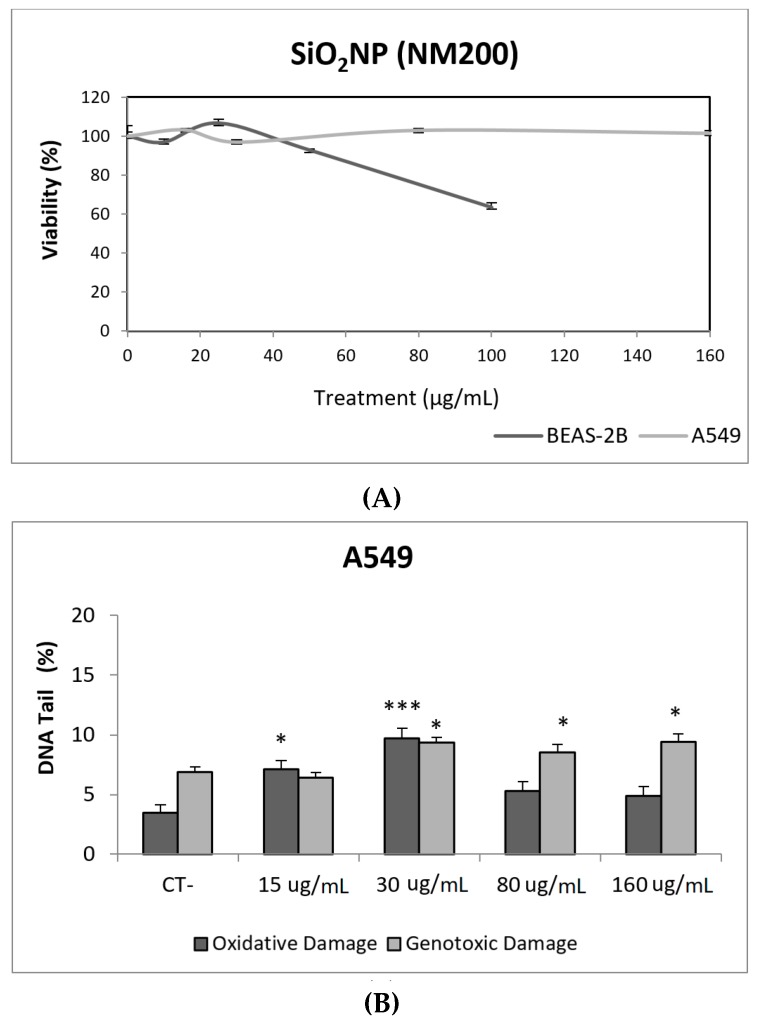
Results from the SiO_2_NPs (NM200), on A549 and BEAS-2B cells. (**A**) Toxicity curves for both cell lines. (**B**) Genotoxicity results obtained in A549 cells. (**C**) Genotoxicity results obtained in BEAS-2B cells. Genotoxicity data are plotted as mean ± SEM of two independent experiments. * *p* ≤ 0.05, *** *p* ≤ 0.001 (one way-ANOVA). CT, control.

**Figure 6 nanomaterials-09-01385-f006:**
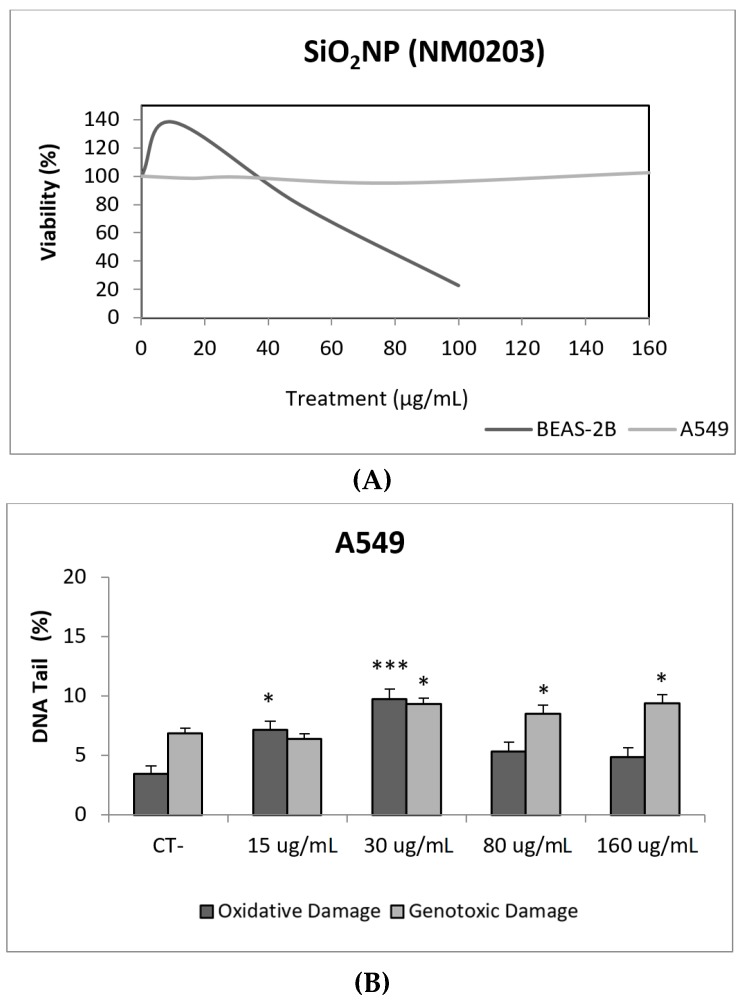
Results from the SiO_2_NPs (NM203), on A549 and BEAS-2B cells. (**A**) Toxicity curves for both cell lines. (**B**) Genotoxicity results obtained in A549 cells. (**C**) Genotoxicity results obtained in BEAS-2B cells. Genotoxicity data are plotted as mean ± SEM of two independent experiments. * *p* ≤ 0.05, *** *p* ≤ 0.001 (one way-ANOVA). CT, control.

**Figure 7 nanomaterials-09-01385-f007:**
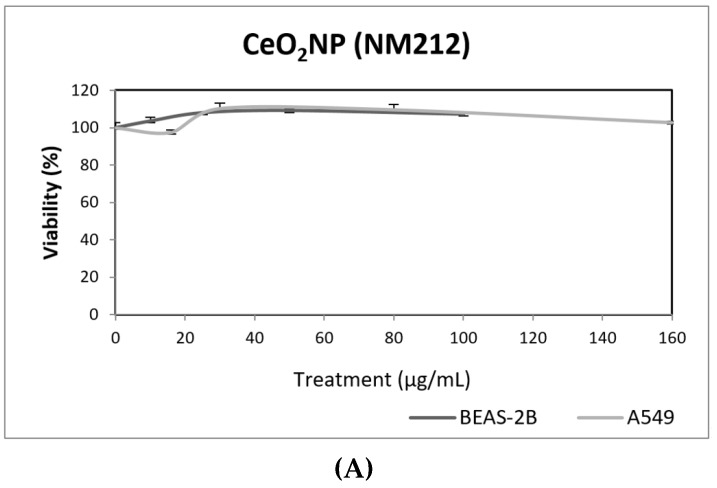
Results from the CeO_2_NPs (NM212), on A549 and BEAS-2B cells. (**A**) Toxicity curves for both cell lines. (**B**) Genotoxicity results obtained in A549 cells. (**C**) Genotoxicity results obtained in BEAS-2B cells. Genotoxicity data are plotted as mean ± SEM of two independent experiments. It should be noted that there is a significant reduction in the levels of oxidative DNA damage observed in the A549 cells. CT, control.

**Figure 8 nanomaterials-09-01385-f008:**
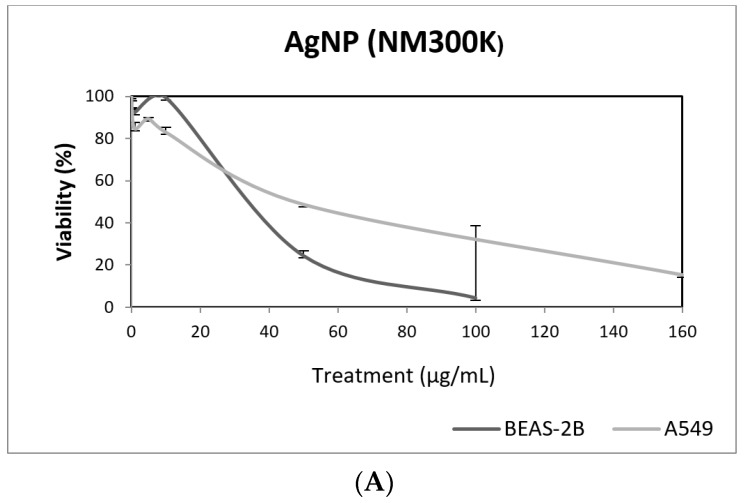
Results from the AgNPs (NM300K), on A549 and BEAS-2B cells. (**A**) Toxicity curves for both cell lines. (**B**) Genotoxicity results obtained in A549 cells. (**C**) Genotoxicity results obtained in BEAS-2B cells. Genotoxicity data are plotted as mean ± SEM of two independent experiments. * *p* ≤ 0.05, ** *p* ≤ 0.01, *** *p* ≤ 0.001 (one way-ANOVA). CT, control.

**Figure 9 nanomaterials-09-01385-f009:**
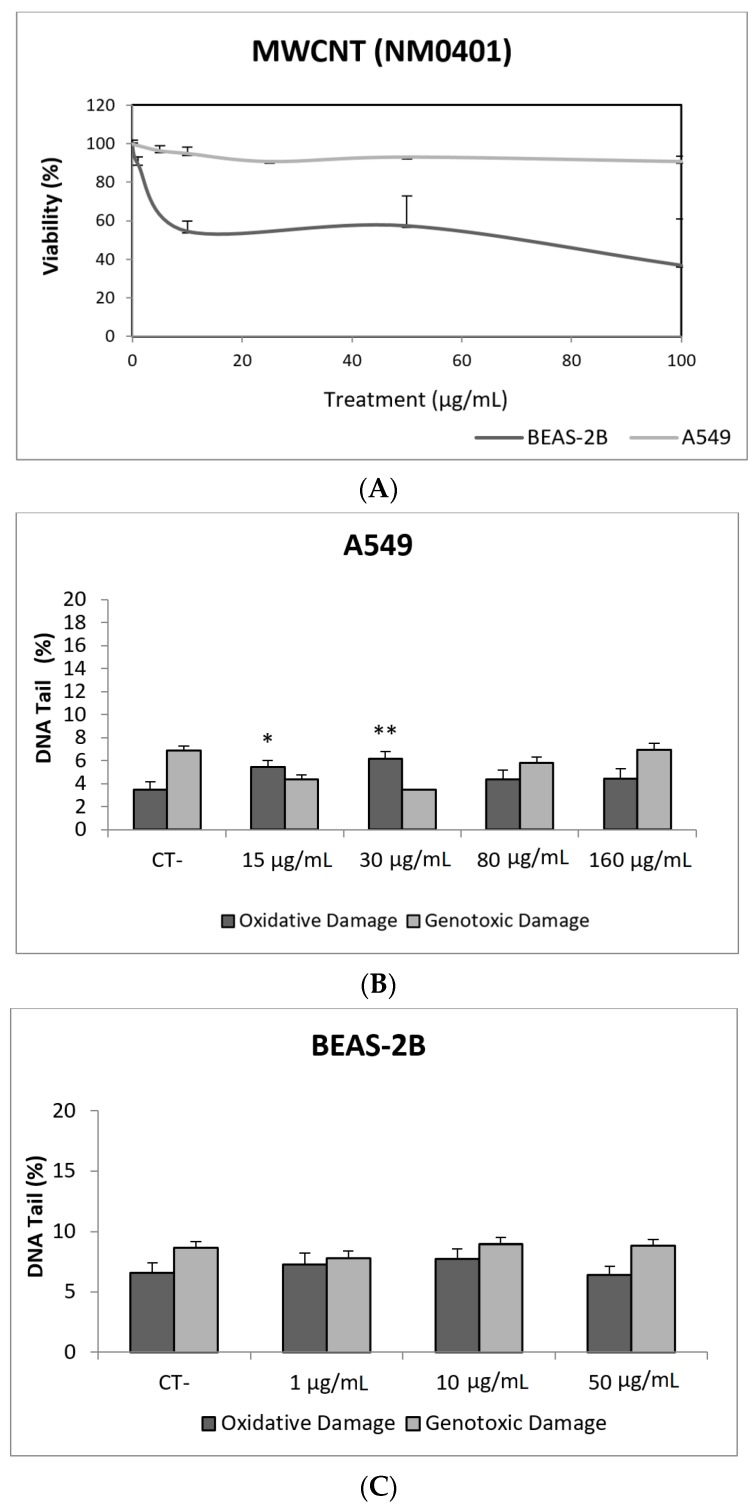
Results from the MWCNTs (NM401), on A549 and BEAS-2B cells. (**A**) Toxicity curves for both cell lines. (**B**) Genotoxicity results obtained in A549 cells. (**C**) Genotoxicity results obtained in BEAS-2B cells. Genotoxicity data are plotted as mean ± SEM of two independent experiments. * *p* ≤ 0.05, ** *p* ≤ 0.01 (one way-ANOVA). CT, control.

**Figure 10 nanomaterials-09-01385-f010:**
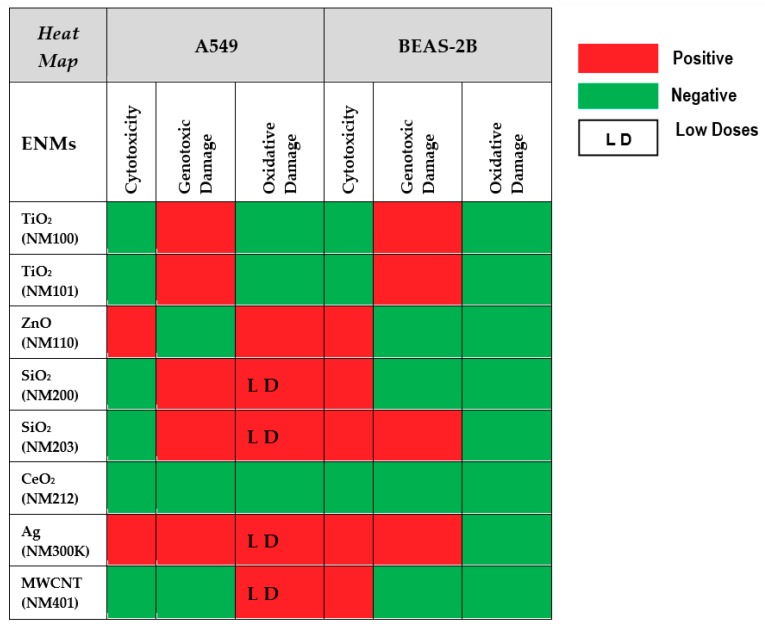
Heat map summarizing the overall results of testing the eight nanomaterials in the two cell lines. Toxicity, genotoxicity and oxidative damage induction are schematically represented.

**Table 1 nanomaterials-09-01385-t001:** Comparison of pristine sizes (transmission electron microscopy, TEM) and 10 and 100 µL dispersions (dynamic light scattering, DLS). Sizes were evaluated just after the dispersion procedure and 24 h later.

NM100 (TiO_2_NPs)				NM101 (TiO_2_NPs)			
TEM	104.01 ± 39.42 nm			TEM	54.69 ± 35.39 nm		
DLS	0 h	10 µg/mL	194.6 ± 8.49	DLS	0 h	10 µg/mL	152,2 ± 62.47
		100 µg/mL	166.2 ± 1.01			100 µg/mL	166.1 ± 12.55
	24 h	10 µg/mL	151.6 ± 6.30		24 h	10 µg/mL	265.1 ± 10.50
		100 µg/mL	141.2 ± 2.23			100 µg/mL	141.5 ± 84.88
NM110 (ZnONPs)				NM200 (SiO_2_NPs)			
TEM	132.37 ± 69.53 nm			TEM	16.5 ± 4.18 nm		
DLS	0 h	10 µg/mL	260.3 ± 88.63	DLS	0 h	10 µg/mL	67.76 ± 0.10
		100 µg/mL	213.9 ± 92.48			100 µg/mL	70.59 ± 1.43
	24 h	10 µg/mL	117.2 ± 37.71		24 h	10 µg/mL	75.84 ± 3.62
		100 µg/mL	114.7 ± 0.86			100 µg/mL	147.8 ± 13.70
NM203 (SiO_2_NPs)				NM212 (CeO_2_NPs)			
TEM	24.26 ± 9.38 nm			TEM	70.33 ± 49.61 nm		
DLS	0 h	10 µg/mL	69.56 ± 9.50	DLS	0 h	10 µg/mL	229.9 ± 10.13
		100 µg/mL	86.86 ± 68.08			100 µg/mL	230.5 ± 4.05
	24 h	10 µg/mL	121.4 ± 10.02		24 h	10 µg/mL	117.1 ± 4.70
		100 µg/mL	342 ± 76.05			100 µg/mL	124.0 ± 1.51
NM300K (AgNPs)				NM401 (MWCNTs)			
TEM	7.75 ± 2.48 nm			TEM	4.23 ± 1.01 (diameter), 6012.09 ± 4091.45 (length)		
DLS	0 h	10 µg/mL	28.71 ± 17.67	DLS	0 h	10 µg/mL	ND
		100 µg/mL	38.46 ± 16.31			100 µg/mL	ND
	24 h	10 µg/mL	81.37 ± 4.63		24 h	10 µg/mL	ND
		100 µg/mL	97.23 ± 6.22			100 µg/mL	ND

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
