# Peer review of "The Comet Assay as a Tool to Detect the Genotoxic Potential of Nanomaterials"

_nanomaterials, 2019, doi:10.3390/nano9101385_

Round 1

Reviewer 1 Report

This revised version of the manuscript has not changed compared with the original one. Therefore, all the previous comments of this reviewer are still to be solved

1) The paper needs a comparative assay to determine whether the comet assay is appropriate or not (there are several other options and an alternative should have been chosen)

2) The cells used for this paper are lung-related but the nanoparticles are of course added in suspension, which would not be the case in the supposed case of inhalation, in my opinion it does not have any sense to use this but to use epithelial cells would be more appropriate.

3) The TEM images are not clear to see the particles, most of them need some closer images and to see if the particles are porous or not

4) Some of the DLS studies have no sense, the agglomeration is increasing with time in some particles and decreasing in some others, the behaviour should be explained in the manuscript

5) Four concentrations (especially only to 100 microg/mL) are not enough for this kind of studies, it is generally agreed from other papers that very low quantities of nanoparticles may also result in angiogenesis and some very low concentrations have not been tested.

6) The table with the green and red colours is fine, but once you think this is the way the nanoparticles are acting, why not to analyze those more hurting and determine some insights of their mechanism of action?

I recommend rejection.

Author Response

Manuscript ID: nanomaterials-538894

Title: The comet assay as a tool to detect the genotoxic potential of nanomaterials

The reviewer consider that our previous answers are not convincing enough and present again the same objections.
We have answered to this reviewer reaffirming our previous arguments and extending the previous comments. In addition, we have add some of these points in the new version of the manuscript. In particular we have emphasized why we used the two epithelial lung cells, and the advantages of the used protocol regarding its use as a throughput approach (when GBF are used) and to detect mechanism of action (when enzymes like FPG are used).
We hope that with the modifications, made following the comments/questions raised by the reviewer, the current version of our paper will be acceptable for publication.

With the new version, we enclosed a marked copy where the modifications are highlighted.

Reviewer 2 Report

the revised MS can be considered for publication.

Author Response

Manuscript ID: nanomaterials-538894

Title: The comet assay as a tool to detect the genotoxic potential of nanomaterials

Authors: Alba García-Rodríguez, Laura Rubio, Laura Vila, Noel Xamena, Antonia Velázquez, Ricard Marcos*, Alba Hernández*

 RESPONSE FROM THE EDITOR

- Minor revision

RESPONSE TO REVIEWERS

-We thank reviewers 1 and 2 who consider acceptable our final version

Reviewer 3 Report

The author's responses to this reviewer are ok and exhaustive 

Author Response

A word document has been added

Round 2

Reviewer 1 Report

The authors did some of the required revisions. I finally recommend acceptance.

This manuscript is a resubmission of an earlier submission. The following is a list of the peer review reports and author responses from that submission.

Round 1

Reviewer 1 Report

This paper is reporting on the cytotoxicity and genotoxicity of nanoparticles of different nature using comet assays. The paper is interesting but it is not appropriate for publication in Nanomaterials at least in its present form.

The paper has two major mistakes in its design:

1) It needs a comparative assay to determine whether the comet assay is appropriate or not (there are several other options and an alternative should have been chosen)

2) The cells used for this paper are lung-related but the nanoparticles are of course added in suspension, which would not be the case in the supposed case of inhalation, in my opinion it does not have any sense to use this but to use epithelial cells would be more appropriate.

Even though the paper presents these problems I would also like to highlight the following weaknesses of the manuscript

1) The TEM images are not clear to see the particles, most of them need some closer images and to see if the particles are porous or not

2) Some of the DLS studies have no sense, the agglomeration is increasing with time in some particles and decreasing in some others, the behaviour should be explained in the manuscript

3) Four concentrations (especially only to 100 microg/mL) are not enough for this kind of studies, it is generally agreed from other papers that very low quantities of nanoparticles may also result in angiogenesis and some very low concentrations have not been tested.

4) The table with the green and red colours is fine, but once you think this is the way the nanoparticles are acting, why not to analyze those more hurting and determine some insights of their mechanism of action?

I recommend rejection in its present form and if the authors are prepared to carry out the proposed studies, I would then reconsider my decision as the paper has potential to be published in Nanomaterials in the future.

Reviewer 2 Report

The authors present cyto- and genotoxicity data about a variety of NM. Instead of evaluating the comet assay as a controversially discussed method to evaluate nanogenotoxicity they just present their results using this single standard method. Addressing the value of the comet assay as a diagnostic tool in nanotoxicology it would have been necessary to compare it to other test methods (micronucleus testing, chromosomal testing). The presented experiments have been performed and published several times before, so I do not really see the novelty of the current work. Furthermore, two airway cell lines were used. These should have been cultivated under air-lift conditions in order to be eligable for toxicity testing. NM characterization is lacking, no ion concentrations of Zn2+ were measured, no TEM pictures of NM uptake into the cells, and so on. 

It would have been interesting to compare different genotoxicity assays in a certain number of NM at the same concentrations. This might have answered to "question" of the headline.

Additional experiments will be needed in order to make the MS publishable 

Reviewer 3 Report

The tested NMs were between the most emerging  TiO2NPs (NM100 and NM101),  SiO2NPs (NM200 and NM203),  nm ZnONPs (NM110), ),  CeO2NPs 64 (NM212), AgNPs (NM300K), and MWCNT (NM401).  

Results indicate that most of the selected nanomaterials showed mild to significant genotoxic effects, at list in the A549 cell line. The work reinforces and confirms the use of the comet assay as a good  tool to detect the genotoxic potential. 

Minor revisions:

lines 178-179: Change Di Bucchiano et al. using NM100 and NM101 compounds (as in our

study) demonstrated a weak but positive induction of genotoxic effects, NM100 being the most

dangerous [20] ..in Di Bucchianico et al....

2. regarding A549 vulnerability: some further comments and references can be added about their genetic instability as cancer cells

3. The genotoxic damage as in figures of A549 in general is not clearly dose dependent: can an explanation be given?